# A Versatile Control Method for Multi-Agricultural Machine Cooperative Steering Applicable to Two Steering Modes

Weizhen Zhu, Yuhao Zhang, Weiwei Kong *, Fachao Jiang and Pengxiao Ji

College of Engineering, China Agricultural University, Beijing 100083, China; sy20223071562@cau.edu.cn (W.Z.); davidzhanghao@me.com (Y.Z.); jfachao@cau.edu.cn (F.J.); pengxiaoji@cau.edu.cn (P.J.)

* Correspondence: kongweiwei@cau.edu.cn

**Abstract:** This article aims to address the unnecessary stopping and low efficiency issues present in existing multi-machine cooperative steering control methods. To tackle this challenge, a novel cooperative control approach for multiple agricultural machines is proposed, considering two typical steering modes of farm machinery. This approach encompasses a multi-machine cooperative control framework suitable for both steering modes. Based on the established lateral and longitudinal kinematics models of the farm machines, the method includes a path-tracking controller designed using the pure pursuit and Stanley algorithms, a formation-keeping controller based on PID control, and a T-turn cooperative-steering controller based on a problem-solving approach. To assess the method's viability, a collaborative simulation platform utilizing CarSim and Simulink was constructed, which conducted simulations for both U-turn and T-turn cooperative steering controls. The simulation results indicate that the proposed control framework and methodology can effectively ensure no collision risk during the U-turn and T-turn cooperative steering processes for three farm machines, eliminating stopping in T-turn, enhancing safety, and improving fuel economy. Compared with traditional sequential control methods, the proposed approach reduced operation time by 17.47 s and increased efficiency by 15.29% in the same scenarios.

**Keywords:** agricultural vehicle steering mode; multiple agricultural machine cooperative control; path tracking; formation maintenance

## 1. Introduction

With the intensification, scaling, and industrialization of agriculture in China, as well as the rising demands for efficiency and the increasing complexity of tasks in agricultural operations, traditional automatic navigation techniques for agricultural machinery are no longer sufficient to meet the challenges of safety and break through efficiency bottlenecks [1]. Therefore, the development of intelligent agricultural machinery technology, including multi-machine cooperative systems, has become an urgent need for the evolution of China's agriculture [2–4].

The multi-machine cooperative technology has garnered extensive recognition from both domestic and international researchers. Its primary components encompass positioning and communication technology, path planning technology, and control technology [5]. The positioning and communication sector is responsible for acquiring and exchanging critical data such as the position and heading of both the subject vehicle and others in the vicinity. To this end, Zhu Z et al. equipped farm machinery with potentiometers, magnetic speed sensors, and RTK-GPS to ascertain front wheel steering angles, velocities, heading angles, and positional coordinates [6]. Gerasimos G. Rigatos proposed a multi-machine collaborative control approach based on state estimation. This method integrates information from multiple sensors using a derivative-free nonlinear filtering technique. The accurate estimation of the agricultural machinery's position and its motion characteristics is achieved by replacing the extended information filter with a recursive standard information filter

for distributed state estimation [7]. Li S et al. designed communication protocol frames for inter-vehicle communication within a master–slave coordinated system among agricultural vehicles, enabling the exchange of status information [8]. Wenju Mao et al. developed a robotic navigation system for orchard-harvesting machines featuring a dual master–slave mode. The communication protocol data frame format between orchard-harvesting robots was elucidated for the transmission of GNSS coordinates, velocity, navigation modes, and other information pertaining to the harvesting and transport machines [9]. Path planning technology considers the optimal path for a multi-machine system to avoid collisions, planning and calculating the route for each agricultural machine accordingly. Martin Andreas Falk Jensen et al. incorporated optimization criteria such as time and operational distance, utilizing the Dijkstra algorithm to address path planning for a multi-machine system in field and intra-field transportation scenarios [10]. Addressing the challenges of heterogeneous agricultural machinery fleets, which vary in speed, fuel consumption, turning radius, and fuel tank size, Jesus Conesa-Muñoz et al. proposed a practical multi-machine cooperative path planning method. This approach deliberates over various optimization standards such as distance, time, and input costs, ensuring a comprehensive strategy for efficient path planning [11]. Timo Blender et al. employed a centralized entity, Opti-Visor, to manage the path planning and optimization for a fleet of agricultural machines. The architecture and functionality of Opti-Visor within the agricultural machine fleet system were delineated [12]. Control technology ensures the synchronized operation of a multi-machine cooperative system in prescribed formations, at designated speeds, and with appropriate longitudinal and lateral spacing as per mission specifications. A cooperative navigation control method for a harvesting fleet, premised upon a leader–follower structure, was proposed by Bai X et al. [13]. To manage headland turning maneuvers, Zhang C et al. devised control strategies and conducted field tests with real machinery, thereby laying the groundwork for operational efficacy in agricultural tasks requiring such complex turns [14].

Current research on multi-machine cooperative localization, communication technology, and path planning has reached a relatively mature stage. However, in the domain of control technology research, most studies are focused on employing various methodologies to address the challenges associated with the collaborative control of two agricultural machines. Noboru Noguchi proposed two fundamental motion control algorithms for master–slave agricultural machine systems: the GOTO algorithm and the FOLLOW algorithm [15]. In response to uncertainties such as variations in workload and soil conditions during agricultural operations, leading to issues of instability, delayed response, and control challenges in the cooperative following of agricultural machinery, Xu Guangfei et al. proposed a hierarchical control architecture for master–slave multi-agricultural machine following [16]. Wang Zhiqing designed the speed control system, heading–following control system, and following distance control system for two agricultural machines, as well as a collaborative control system for the lead and follow vehicles [17]. Chi Zhang et al. proposed a multi-tractor system for field operations aimed at reducing total operation time, improving work efficiency, and presenting strategies for lateral and longitudinal spacing control [18]. While a minority of studies address three or more agricultural machines, Stavros G. Vougioukas proposed a distributed control framework for multiple agricultural machines, wherein each machine was equipped with a nonlinear model predictive tracking controller. Additionally, it received motion trajectories from other machines, incorporating them into its own control considerations [19]. Zhang Wenyu et al. designed a longitudinal relative position collaborative control method suitable for master–slave navigation in coordinated harvesting and unloading operations [20]. Zheng Xinyao addressed the steering control issue for the follower machine by proposing a fuzzy adaptive PID variable damping steering control method. This approach aims to ensure precise and rapid tracking of the desired steering angle command for the follower machine's steering wheel [21]. Gou ABE et al. developed a control algorithm based on a laser scanner to recognize the relative distance and direction to the lead vehicle and follow it [22].

In summary, existing research on multi-machine cooperative control is limited to linear path coordination, and the rare studies on headland turn control typically focus on only one type of turning method and often employ time-sequential control [23,24]. This approach can lead to unnecessary stopping and waiting, reducing operational efficiency. The main contributions of the paper are as follows:

(1) This article proposes a versatile multi-machine control architecture, applicable to two typical agricultural machine steering modes, including U-turn and T-turn.
(2) Building upon the control architecture, a cooperative control approach for multiple agricultural machines in both straight-line and headland turn steering scenarios is proposed. This method, while ensuring the safety of agricultural machines, aims to enhance operational efficiency and fuel economy.

The rest of the paper is organized as follows: Section 2 provides a detailed description of the problem addressed in this paper, while Section 3 presents the overall control architecture, establishes the longitudinal and lateral kinematic models for agricultural machine, and designs a multi-robot cooperative controller based on this foundation. Section 4 discusses the simulation results; Section 5 concludes the article and discusses future work.

## 2. Problem Description

This article focuses on a cohort of three homogenized machines equipped with implements, aimed at maintaining formation during straight-line operations, as well as ensuring safety and efficiency during headland turns. The lead machine in this group is driven by a human, operating at a constant speed, while the second and third tractors are autonomous agricultural machines, programmed to follow their respective preceding vehicle.

During straight-line operations, the following agricultural machines maintain the formation, as depicted in Figure 1, while working in tandem with the lead agricultural machine. The green lines in the Figure 1 refer to the operation lines of agricultural machine.

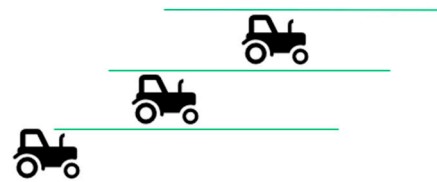

**Figure 1.** Multi-machine straight-line operation formation.

During headland turns in agricultural operations, the steering method of the machinery can be classified into U-turn and T-turn, based on the relationship between the machine's minimum turning radius ($r$) and the working width ($w$), as shown in Figure 2. The turning path consists of two terminal circular arcs, $T_1T_2$ and $T_3T_4$, and a connecting line segment, $T_2T_3$. $T_1$ marks the commencement of the turn, while $T_4$ indicates the completion. Centers $O_1$ and $O_2$ correspond to the turning circular arcs $T_1T_2$ and $T_3T_4$, respectively, with $r$ representing the turning radius, and $w$ indicating the working width. A U-turn approach is used when the working width $w$ is greater than or equal to twice the turning radius, signified as $w \geq 2r$; conversely, a T-turn method is preferred when the working width $w$ is less than double the turning radius, denoted as $w < 2r$ [25].

Addressing straight-line conditions, control measures are implemented on the agricultural machines to maintain a designated formation among the three machines while ensuring operational safety. Concerning headland turn coordination, the aim is to minimize the overall turning time without any risk of collision, and to eliminate stopping and waiting during the turn, thereby enhancing turning efficiency and fuel economy.

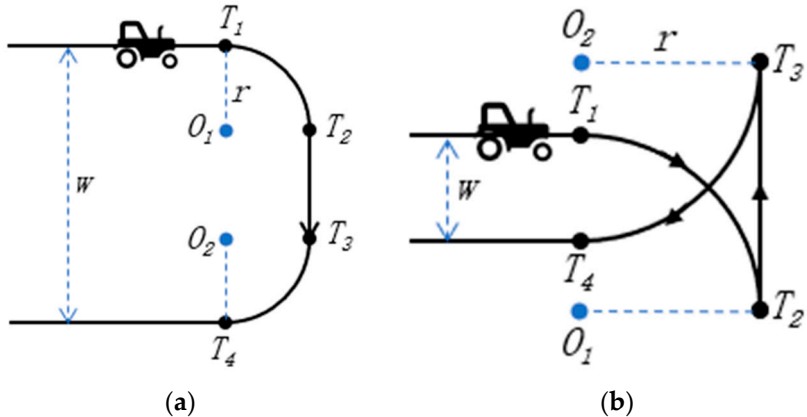

**Figure 2.** Agricultural machine steering model: (**a**) U-turn and (**b**) T-turn.

### 3. Methodology

*3.1. Integral Control Framework*

The multi-machine cooperative control architecture proposed in this study primarily comprises a comprehensive kinematic model for universal agricultural machinery, a path-tracking controller, a formation-maintenance controller, a collision risk detection module, and a controller to optimize cooperative steering during T-turns. This framework ensures that the steering actions performed during U-turns do not elevate the risk of collision; thus, an additional controller for cooperative steering optimization during U-turns is unnecessary. It is sufficient to confirm the safety of the agricultural machinery throughout these maneuvers.

The kinematic model of agricultural machinery, encompassing both lateral and longitudinal dynamics, forms the essential groundwork for the associated lateral and longitudinal control systems. A path-tracking controller is derived based on the lateral kinematic model of the agricultural machinery, enabling machinery operation along pre-planned straight and headland turn paths. Furthermore, based on the longitudinal dynamic model and data such as speed and spacing from the onboard millimeter-wave radar, a multi-machine formation-keeping controller for straight trajectories is designed. This controller generates the necessary speeds to maintain formation, which serves as the speed input for the path-tracking controller on straight paths.

Additionally, a collision risk detection module based on the Separating Axis Theorem was designed to accommodate multi-machine coordinated steering under two steering modes. This module verifies the safety of three agricultural machines during the cooperative steering, providing collision-free minimum spacing for the formation-keeping controllers of both steering types. For the multi-machine T-turn cooperative steering, a steering optimization controller was engineered. Its output determines the speed under the cooperative steering path of the path-tracking controller, enabling three agricultural machines to execute the cooperative steering without topping and waiting, thus minimizing the overall turning time. The integrated control architecture is illustrated in Figure 3, and the control flow chart is shown in Figure 4. $v$ and $\delta$ in Figure 4 represent the desired speed and the desired front wheel angle that the formation-keeping controller and pure pursuit controller outputs to the kinematic model.

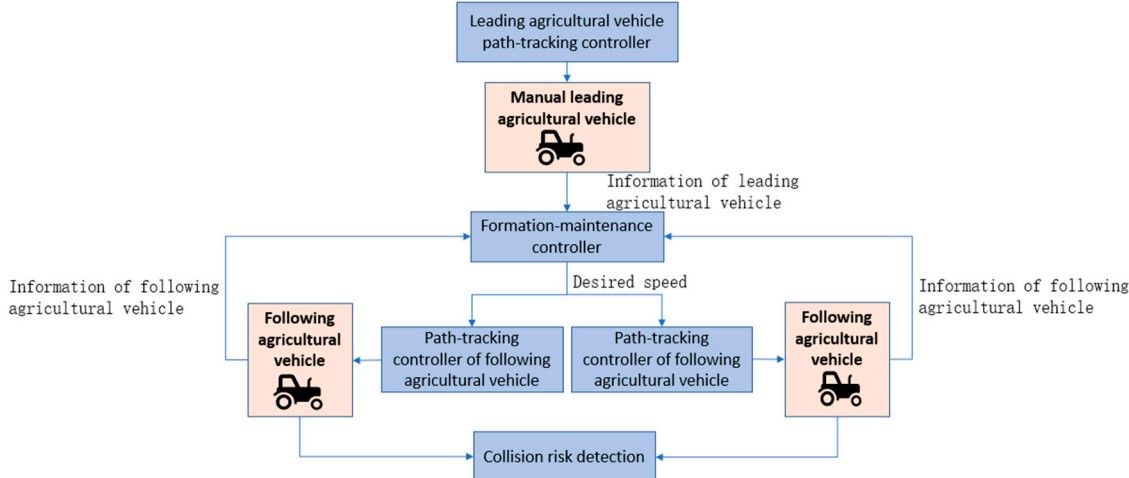

**Figure 3.** Integrated control architecture.

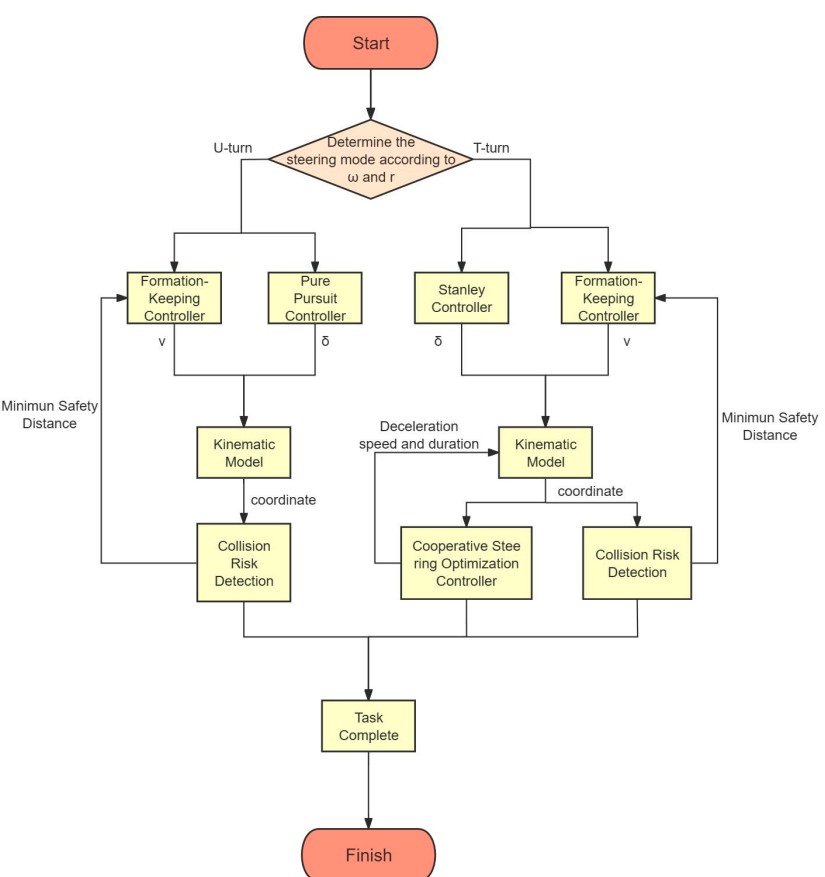

**Figure 4.** Control flow chart.

### 3.2. Kinematic Model

3.2.1. Lateral Kinematics

The lateral kinematic model provides a basis for the design of the path-tracking controller. Assuming there is no interaction between the machine and the ground, and disregarding motions such as roll, pitch, and sideslip, the agricultural machinery can be simplified to a two-wheeled machine model for kinematic analysis, as illustrated in Figure 5. Within this simplified model, point M represents the point on the path that is closest to the control point O.

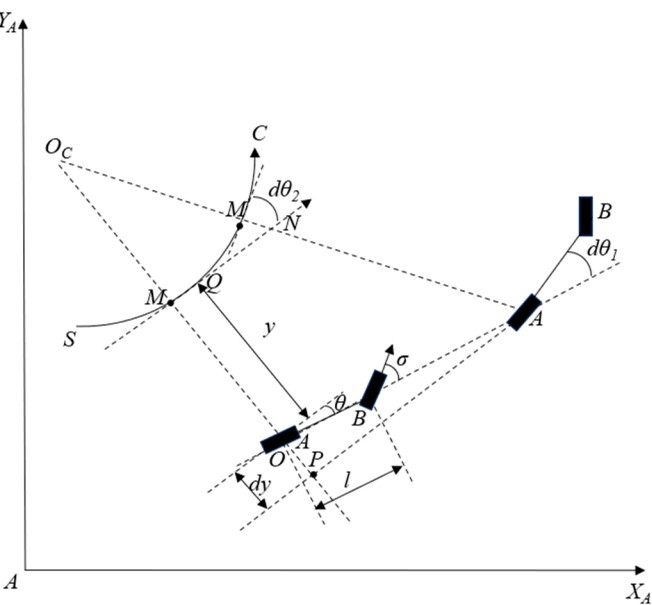

**Figure 5.** Lateral kinematics model diagram.

Based on the schematic of the kinematic model demonstrated above, along with geometric knowledge and physical principles, it is possible to derive a set of differential equations representing the kinematics of agricultural machinery, which are presented as follows:

$$
\begin{cases}
\frac{ds}{dt} = \frac{v\cos\theta}{1-c(s)y} \\
\frac{dy}{dt} = v\sin\theta \\
\frac{d\theta}{dt} = v\left(\frac{\tan\delta}{l} - \frac{c(s)\cos\theta}{c(s)y}\right)
\end{cases}
\tag{1}
$$

where $c(s)$ is the curvature of the following curve path at point M; $v$ is the speed of the agricultural machine; $l$ is the wheelbase of the agricultural machine; $s$ is the arc length of point M along curve C; $y$ is the lateral deviation of the agricultural machine from the set path; $\theta$ is the heading deviation angle of the agricultural machine; and $\delta$ is the steering wheel deviation angle of the agricultural machine.

The kinematic model for agricultural machinery along a straight-path trajectory is described as follows:

$$
\begin{cases}
\frac{ds}{dt} = v\cos\theta \\
\frac{dy}{dt} = v\sin\theta \\
\frac{d\theta}{dt} = v\frac{\tan\delta}{l}
\end{cases}
\tag{2}
$$

Performing elementary calculations yields the system of differential equations for the lateral kinematic model of the agricultural machine, as follows:

$$
\begin{cases}
\frac{dy}{ds} = \tan\theta \\
\frac{d\theta}{ds} = \frac{\tan\delta}{l\cos\theta}
\end{cases}
\tag{3}
$$

### 3.2.2. Longitudinal Kinematics

The longitudinal kinematic model provides a foundation for the design of formation-maintenance controllers. The longitudinal motion state of the following agricultural machinery during cooperative operation can be represented as:

$$
\frac{N_g N_0 \eta_t}{r_w} M_e - K_P P_b = m\dot{v}_f + C_A v_f^2 + mgf
\tag{4}
$$

where $N_g$ is the engine power transmission ratio; $N_0$ is the engine-to-wheel transmission ratio; $\eta_t$ is the mechanical efficiency of the transmission system; $M_e$ is the engine output

torque; $K_p$ is the scaling factor; $P_b$ is the brake pressure; $r_w$ is the wheel radius; $v_f$ is the speed of the following agricultural machine; m is the weight of the following agricultural machine; $C_A$ is the aerodynamic drag coefficient; $g$ is the acceleration due to gravity; and $f$ is the rolling resistance coefficient.

During the following process, a fixed spacing model is employed, from which the desired following distance, denoted as $d_d$, can be derived using the collision risk detection module. For detailed derivation, refer to Section 3.3.2. The derived longitudinal kinematic model for agricultural machinery following is as follows:

$$\Delta \dot{d} = \Delta v \tag{5}$$

$$\Delta \dot{v} = \frac{1}{m}\left(C_A v_f{}^2 + mgf\right) - \frac{N_g N_0 \eta_t}{mr_w} M_e \tag{6}$$

where $\Delta d$ is the following distance error; $d_r$ is the actual following distance; $\Delta v$ is the following speed error; and $v_m$ is the speed of the leading machine.

*3.3. Controller Design*

In accordance with the integral control framework and the kinematic models established in Section 2, the multi-machine cooperative control system is delineated into three distinct components: the linear formation-keeping controller, the headland cooperated steering optimization controller, and the path-tracking controller design. The formation-keeping controller and the headland cooperated steering optimization controller function to furnish the longitudinal velocity inputs requisite for the path-tracking controller, which undertakes the lateral control tasks of the agricultural machinery.

3.3.1. Straight-Line Formation-Keeping Controller

The functionality of the formation-keeping controller lies in adjustments made to the following farm machine's speed, ensuring that the follower maintains a set longitudinal distance from the leader during field operations.

The controller employs a fixed-spacing strategy PID controller, in which the spacing error '$e_i$' is fed into the PID as an error term. This error is then subjected to a linear combination of proportional, integral, and differential calculations to produce the control variable. The model is illustrated in Figure 6.

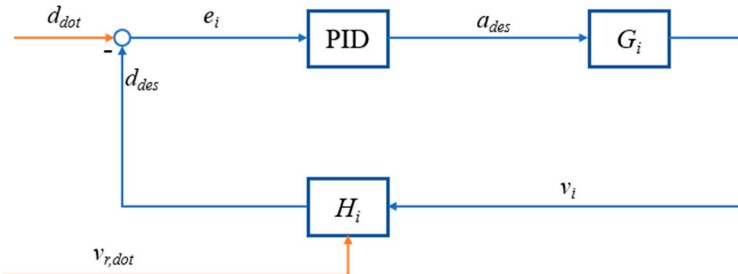

**Figure 6.** PID controller model.

After being processed by the PID controller, the error '$e_i$' yields the desired acceleration for the subject machine. The module '$H_i$' represents the spacing strategy, which determines the desired spacing between the subject machine and the lead machine. The '$G_i$' module is the longitudinal machine controller, which provides the subject machine's velocity as the input to the spacing strategy module.

### 3.3.2. Headland-Turn Cooperative Steering Optimization Controller

(1) Collision Risk Detection

A safety model for agricultural machinery is structured through the implementation of an Oriented Bounding Box (OBB) [26] to identify zones susceptible to collision. Given that the outline of the agricultural machine typically projects a nearly rectangular shape on the ground, a rectangle defined by the points $P_1 P_2 P_3 P_4$ serves as the enclosing surface of the safety model, as illustrated in Figure 7. The dimensions of this rectangle are the combined aggregate of the length and width of the machine's ground projection including its implements, with an additional minimum safety margin incorporated. The heading angle of the machine is represented by $\theta$, and a minimum safety distance of 0.5 m is established to uphold the standard of operational safety.

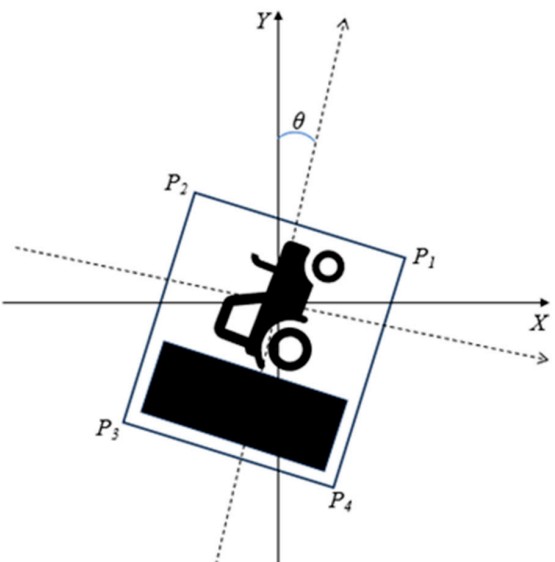

**Figure 7.** Agricultural machine safety model.

Utilizing the Separating Axis Theorem [27], a model for detecting the safety state of agricultural machinery is established, as illustrated in Figure 8. In this model, $O_A$ and $O_B$ are identified as the geometric centers of the OBB for machine 1 and 2, respectively.

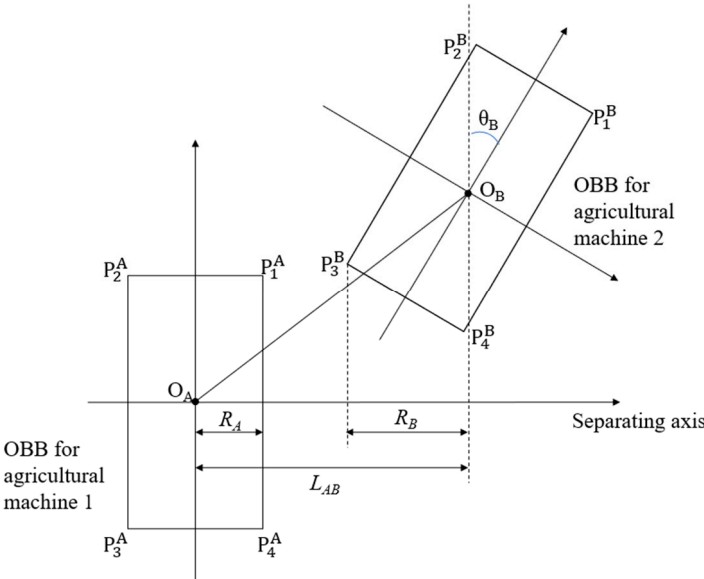

**Figure 8.** Agricultural machine safety state detection mode.

A separated axis coordinate system is established using $O_A$ as the origin, where the separating axis is parallel to the lateral symmetry axis of the OBB of machine 1. $L_{AB}$ denotes the projection length of the geometric center distance between the OBB of the two machines on the separating axis. $R_A$ and $R_B$ are the half-lengths of the maximum projection lengths on the separating axis of machine 1 and 2's OBB section, which can be referred to as the projection radius. The formulas for calculating $R_A$, $R_B$, and $L_{AB}$ are as follows:

$$\begin{cases} R_A = W_A/2 \\ R_B = \frac{W_B cos\theta_B}{2} + \frac{L_B sin\theta_B}{2} \\ L_{AB} = x_B - x_A \end{cases} \tag{7}$$

where $W_A$ is the width of the OBB for agricultural machine 1; $\theta_B$ is the heading angle of agricultural machine 2 relative to agricultural machine 1; $L_B$ is the length of the OBB for agricultural machine 2; and $W_B$ is the width of agricultural machine 2.

According to the Separating Axis Theorem, if the combined projection radius of the agricultural machine OBB sections on the separating axis (denoted as $L_{AB}$) exceed the sum of the projection lengths of the geometric center distances on the separating axis (denoted as $R_A + R_B$), it indicates that there is no collision between the machines, hence, no risk of collision.

(2)    U-turn Cooperative Steering

As the agricultural machinery reaches the headland and begins the coordinated steering of U-turns, the distance between the machines widens. Therefore, to achieve effective control, it is necessary to minimize the spacing as much as possible while ensuring there is no risk of collision during straight-line operations.

(3)    T-turn Cooperative Steering

In T-turn cooperative steering, the large turning radius of the agricultural machine makes it unable to directly turn into the next work path without reversing adjustments. This reduces the distance between machines and increases the risk of collision. Hence, the two following machines must decelerate in advance during straight-line operations to avoid stopping and waiting in the entire cooperative steering process. After the lead agricultural machine and the first following machine have completed the turn, to ensure the distance between the machines is within a safe range, both the lead and the first following agricultural machine need to decelerate. They should continue at a reduced speed until the minimum safe distance is restored, after which they can resume work at the original speed. The control logic is as follows: First, the formation-keeping controller shortens the distance to the minimum safety distance. Once the three machines reach a stable speed, the T-turn headland-steering controller calculates the deceleration, target speeds, and the duration of target speeds for the two following machines, so they can complete the turn without stopping and waiting and without the risk of collision. After completing the turn, the controller instructs the lead agricultural machine and the first following machine to decelerate until they reach the minimum safe distance, after which they resume working speed to continue operations.

Based on the above description, the design of the headland steering optimization controller is as follows:

$$S_{ahead} - S_{ego} \geq L \tag{8}$$

where:

$$S_{ahead} = \int_0^{t_2} v_{ahead} dt \tag{9}$$

$$S_{ego} = \int_0^{t_1} (-at + v_{ego}) dt + \int_{t_2}^{t_2} v_{desire} dt \tag{10}$$

$$L = L_{ABmin} tan\theta_B \tag{11}$$

And to ensure the comfort during the deceleration process of the agricultural machines, the acceleration '$a$', which is measured in meters per second squared (m/s$^2$), needs to satisfy:

$$a \leq 1.5 \tag{12}$$

In Equations (9)–(11), $S_{ahead}$ denotes the travel distance of the preceding machine during the deceleration process, measured in meters (m); $S_{ego}$ signifies the travel distance of the ego vehicle, which refers to the machine under consideration or the machine executing the maneuver, also measured in meters (m).; $L_{ABmin}$ is the minimum safe distance for the agricultural machine; $V_{ahead}$ is the lead machine speed; $V_{ego}$ is the machine speed before it decelerates; $V_{desire}$ is the machine speed after it decelerates; $t_1$ is the duration of the deceleration phase; and $t_2$ is the duration of the $V_{desire}$.

Based on the above controllers, the deceleration, target speed, and the duration of the target speed during the T-type coordinated steering process can be determined. These values serve as inputs for the following agricultural machine.

### 3.3.3. Path-Tracking Controller

The primary function of the path-tracking controller is to ascertain that the lead agricultural machine and the two following machines accurately execute their operations along designated pathways, thereby maintaining a preset lateral distance between each machine, equivalent to the working row width of the machines. The controller employs the actual positional data of the agricultural machinery along with the lateral and heading error from the set path as input parameters. Through a carefully designed control algorithm, it generates the desired steering angle for the wheels. This process ensures that the lateral distance between each following machine and the leading machine is consistently maintained within the defined numerical range. The integrity of this system is critical to achieve precise alignment and coordination in agricultural operations, enabling the efficient utilization of the machinery within the prescribed operational parameters.

(1)    U-turn Cooperative Steering

For the comparatively straightforward U-turn cooperative steering, the path-tracking controller utilizes a pure pursuit algorithm, as illustrated in Figure 9. In this schematic, point A represents the center of the rear axle, B is the center of the front axle, and C designates the target point on the path closest to point A.

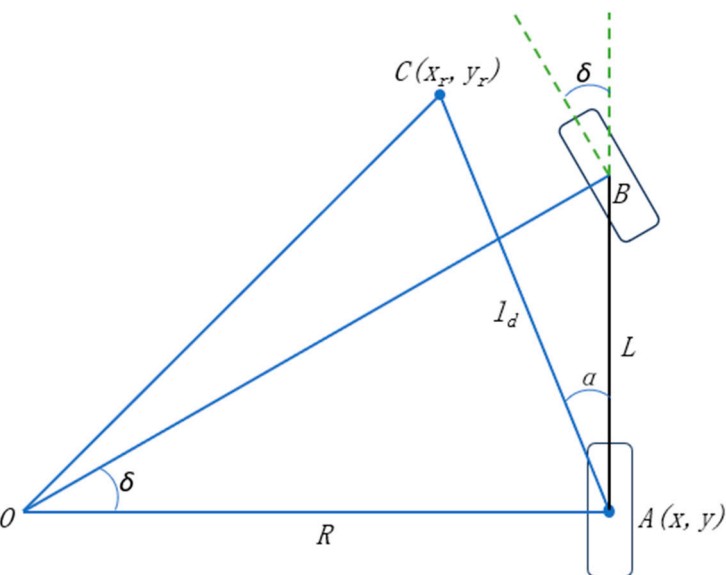

**Figure 9.** Schematic diagram of pure pursuit algorithm.

The geometric relationships in conjunction with the formula for calculating the turning radius yield the following result:

$$\begin{cases} \frac{1}{R} = \frac{2sin\alpha}{l_d} \\ tan\delta = \frac{L}{R} \end{cases} \tag{13}$$

The target steering angle for the front wheels can be derived as follows:

$$\delta = arctan\left(\frac{2Lsin\alpha}{l_d}\right) \tag{14}$$

where $R$ is the steering radius; $L$ is the wheelbase; $\delta$ is the front wheel angle; $\alpha$ is the angle between the machine body and the look-ahead point; and $l_d$ is the look-ahead distance.

'$e$' is defined as the lateral error of the look-ahead point, then

$$e = l_d sin\alpha \tag{15}$$

The control law for the pure pursuit algorithm is derived as follows:

$$\delta = arctan\left(\frac{2Le}{l_d{}^2}\right) \tag{16}$$

(2) T-turn Cooperative Steering

For the T-turn cooperative steering, which involves both forward and reverse movements, the path-tracking controller adopts the Stanley algorithm, which can better handle curved paths and lateral errors. The principles of this algorithm are depicted in Figure 10. Analogous to the pure pursuit algorithm, point A marks the center of the rear axle, and point B marks the center of the front axle. Point P represents the nearest path point to the center of the front axle.

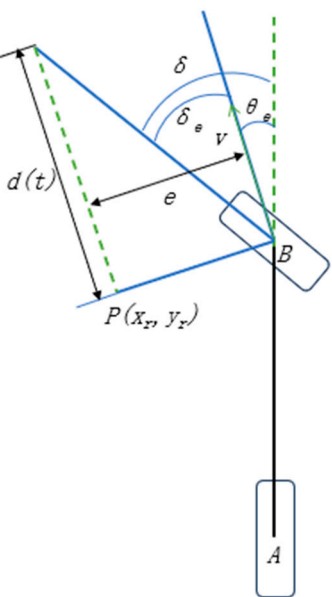

**Figure 10.** Schematic diagram of Stanley algorithm.

Neglecting the lateral tracking error '$e$', the angle of the front wheel aligns with the tangent to the given path, which results in the following equation:

$$\delta_{\theta_e} = \theta_e(t) \tag{17}$$

Disregarding the heading error ($\theta_e$), a more substantial lateral tracking error results in a greater steering angle of the front wheel. Assuming that the vehicle's anticipated

trajectory intersects with the tangent of the nearest point on the given path at a distance $d(t)$ ahead of the front wheel, the following nonlinear proportional function is deduced based on the geometric relationships:

$$\delta_e(t) = arctan\frac{ke(t)}{v(t)} \qquad (18)$$

Considering both control factors, the control law for the front wheel steering angle is as follows:

$$\delta(t) = \theta_e(t) + \frac{ke(t)}{v(t)} \qquad (19)$$

Based on the linear bicycle kinematics model and geometric relationships, the rate of change of lateral error can be obtained as follows:

$$\dot{e}(t) = \frac{-ke(t)}{\sqrt{1+\left(\frac{ke(t)}{v(t)}\right)^2}} \qquad (20)$$

When the lateral tracking error $e(t)$ is small, it can be deduced that:

$$e(t) = e(0) \times e^{-kt} \qquad (21)$$

Therefore, the lateral error exponent converges to $e(t) = 0$, where the parameter $k$ determines the convergence speed of the lateral error. For any lateral error, the differential equation monotonically approaches 0.

## 4. Simulation Validation

To validate the model and method proposed in this article, a simulation scenario was established in CarSim, as shown in Figure 11, and the proposed control algorithm was constructed in Simulink to set up a CarSim/Simulink joint simulation platform. The platform was then used to conduct a simulated validation of U-turn and T-turn cooperative steering control with one lead agricultural machine and two following agricultural machines.

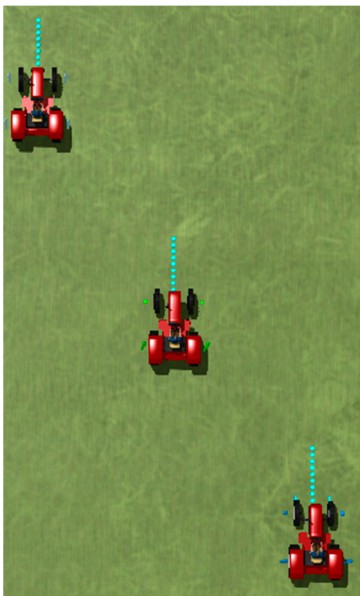

**Figure 11.** CarSim simulation scenario diagram.

### 4.1. Scenario Design

The YTO-LX800 tractor is chosen for this study, and its parameters are presented in Table 1. It is equipped with implements of dimensions 8000 × 2000 mm and 4000 × 2000 mm for U-turn and T-turn scenarios, respectively, determined based on the relationship between turning radius and working width. The safety model can be approximated as a 7 × 7 m rectangle.

**Table 1.** Parameters of YTO-LX800.

| Parameter | | Value |
|---|---|---|
| Model | | YTO-LX800 |
| Dimensions (mm) | | 4250 × 2090 × 2850 |
| Minimum turning radius (m) | | 4 |
| Wheelbase (mm) | | 2342 |
| Total vehicle weight (kg) | | 2725 |
| Speed range (km/h) | Forward | 1.92 to 31.72 |
| | Reverse | 5 to 15.01 |
| Maximum traction force (kN) | | ≥22.4 |

A field block, MNPQ, is selected, approximately 100 m in length, and the routes for three tractors to perform cooperative U-turn and T-turn maneuvers are planned, as shown in Figure 12. Different colors in Figure 12 represent the paths of different farm machines. The path AB represents the path of the lead machine, while $A_1B_1$ and $A_2B_2$ correspond to the paths of the following machines. The coordinates of the path points are entered into the path-tracking controller. It is stipulated that the machines maintain a stable working speed of 10 km/h, while the speed during reverse motion is set to 5 km/h. Furthermore, the acceleration during the entire process is to be confined within the limits of ±1.5 m/s².

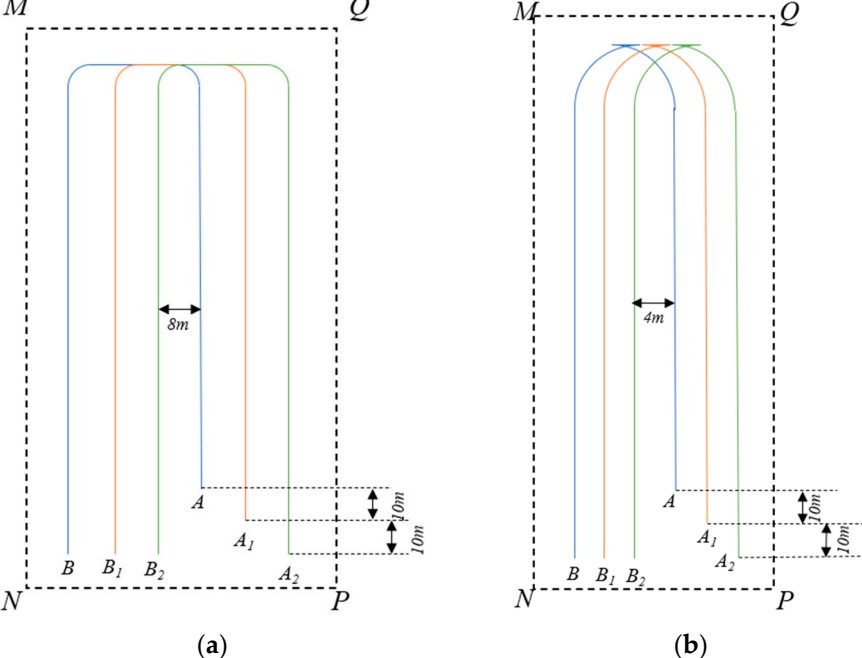

**Figure 12.** Path diagram: (**a**) U-turn cooperative steering and (**b**) T-turn cooperative steering.

In addition, due to the significant curvature changes in the T-turn cooperative steering and the involvement of forward and reverse path transitions, it is necessary to divide the T-turn into three segments and track each segment separately, as shown in Figure 13. Serial number 1–3 in Figure 13 indicates the sequence of the segmented path of agricultural

machine. The switching conditions between different paths are determined based on factors such as the position and heading angle of the tractors.

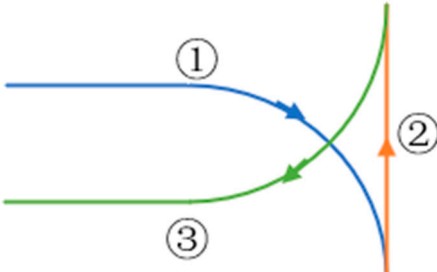

**Figure 13.** Path segmentation diagram.

A joint simulation platform with Carsim and Simulink is built, where U-turn and T-turn cooperative scenarios for straight-line and steering path simulations are constructed within Carsim. Meanwhile, in Simulink, the various controllers, as described in Section 3, are established for simulation validation.

*4.2. Analysis of Results*

4.2.1. U-Type Cooperative Steering

Under the U-turn cooperative steering scenario, the path tracking trajectories of three agricultural machines are shown in Figure 14, where VL represents the lead machine, while VF1 and VF2 are the first and second following machines, respectively. On each machine's trajectory, squares are used to denote the position of the machine at 16 different points in time, with the number 1–16 indicating the sequence of moments. Establishing VF2's initial position as the coordinate origin, with the travel direction along the positive *X*-axis and the steering direction along the positive *Y*-axis, the initial coordinates for VL, VF1, and VF2 are defined as (20, 12), (10, 6), and (0, 0), respectively. The collaborative operation task is deemed completed and the simulation is terminated upon VF2 achieving a new *X*-coordinate of 0 after steering, corresponding to the coordinates (0, 18).

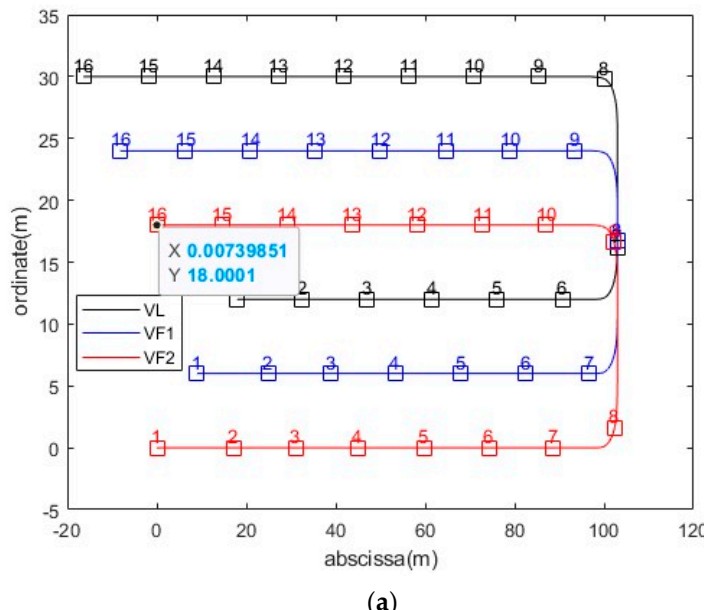

(**a**)

**Figure 14.** *Cont*.

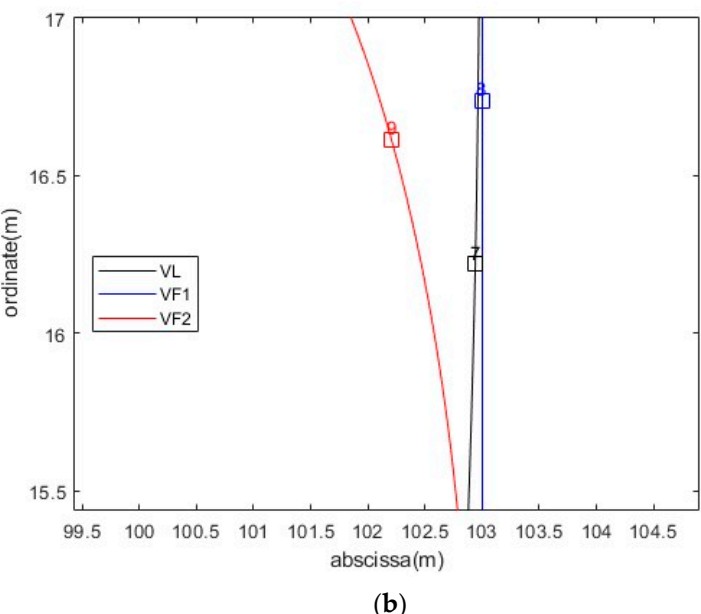

**(b)**

**Figure 14.** U-type cooperative steering trajectory diagram: (**a**) global trajectory and (**b**) local trajectory.

The longitudinal speeds are depicted in Figure 15, where the two following agricultural vehicles initially accelerate to reduce the distance to the leading vehicle and then reach a stable state at approximately 30 s. The simulation concludes at the 78.353 s mark upon the completion of the operational task. The inter-vehicle spacing is illustrated in Figure 16, with VL/VF1 representing the distance between the leading agricultural vehicle and the first following agricultural vehicle, and VF1/VF2 indicating the spacing between the first and second agricultural vehicles. Additionally, the red dashed line in Figure 16 represents the minimum safety distance, and the results from the collision risk detection module consistently indicate the absence of collision risks throughout the entire process.

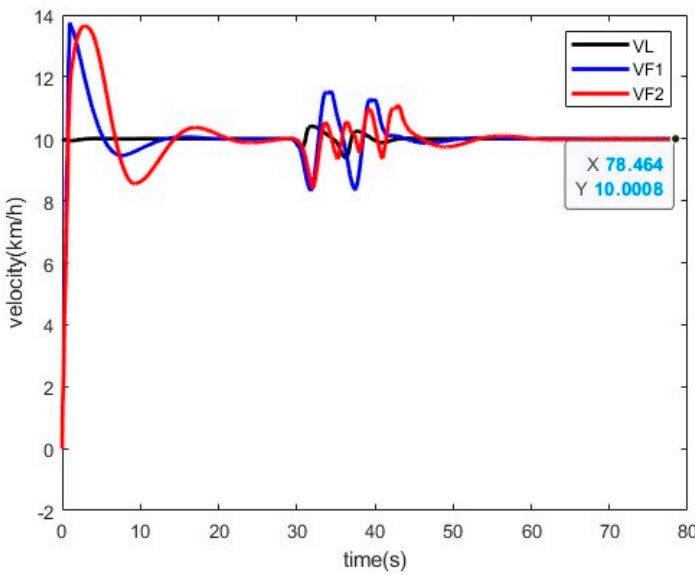

**Figure 15.** U-type cooperative steering longitudinal speed diagram.

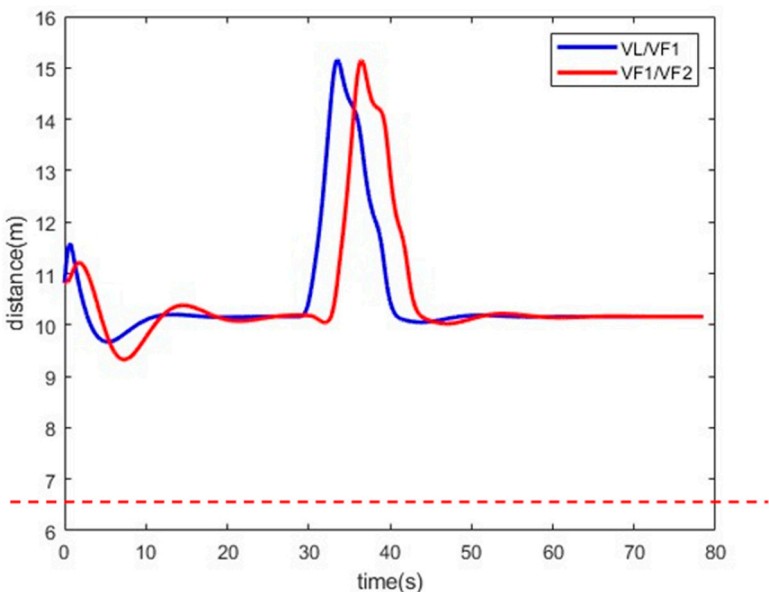

**Figure 16.** U-type cooperative steering agricultural vehicle distance diagram.

Based on the trajectory and spacing charts from the U-turn cooperative steering simulation, as well as the results of the collision risk detection module, it can be discerned that throughout the whole process, the distances between the agricultural machines were always above the minimum safety gap. Moreover, at any given moment, the plotted paths of the machines did not conflict, and there was no risk of collision.

The simulation results indicate that the method proposed in this study for the multiple agricultural machine cooperative operation and U-turn cooperative steering enables the leader and follower machines to work according to the planned paths while maintaining a certain formation. Furthermore, throughout the entire operation and cooperative steering process, there is no risk of collision between the machines.

### 4.2.2. T-Type Cooperative Steering

In a T-turn cooperative steering scenario, the path tracking trajectories of three agricultural machines are depicted in Figure 17. The squares and numerals within the figure carry the same meaning as in the U-turn cooperative steering scenario, representing the location of the machines at various time intervals and the sequential order of these moments, respectively. Similarly, stipulating VF2′s initial position as the coordinate origin, with the travel direction along the positive *X*-axis and the steering direction along the positive *Y*-axis, the initial coordinates for VL, VF1, and VF2 are set at (20, 6), (10, 3), and (0, 0), respectively. The collaborative operation task is considered complete and the simulation is halted upon VF2 reaching a new X-coordinate of 0 after steering, corresponding to the coordinates (0, 9).

As illustrated in Figure 18, the longitudinal speed of the agricultural machinery begins with an acceleration by the two following machines to reduce the distance to the lead vehicle, subsequently stabilizing at 10 km/h. Once stable, the headland cooperative steering controller calculates the required deceleration, target velocity, and the duration for maintaining the target velocity to ensure a collision-free turning process. This results in a controlled deceleration of the two following machines, corresponding to the section between the blue lines in the diagram. In particular, the second following machine must sustain the target speed for a longer period to increase the gap from the first following machine. After the lead agricultural machine and the first trailing machine have completed the turn, they decelerate to wait for the following machine, corresponding to the section between the green lines in the diagram. The three machines then shorten the interval between them to the minimum safety distance before resuming a stable speed and state. At 96.73 s, the operational task is completed, and the simulation is terminated.

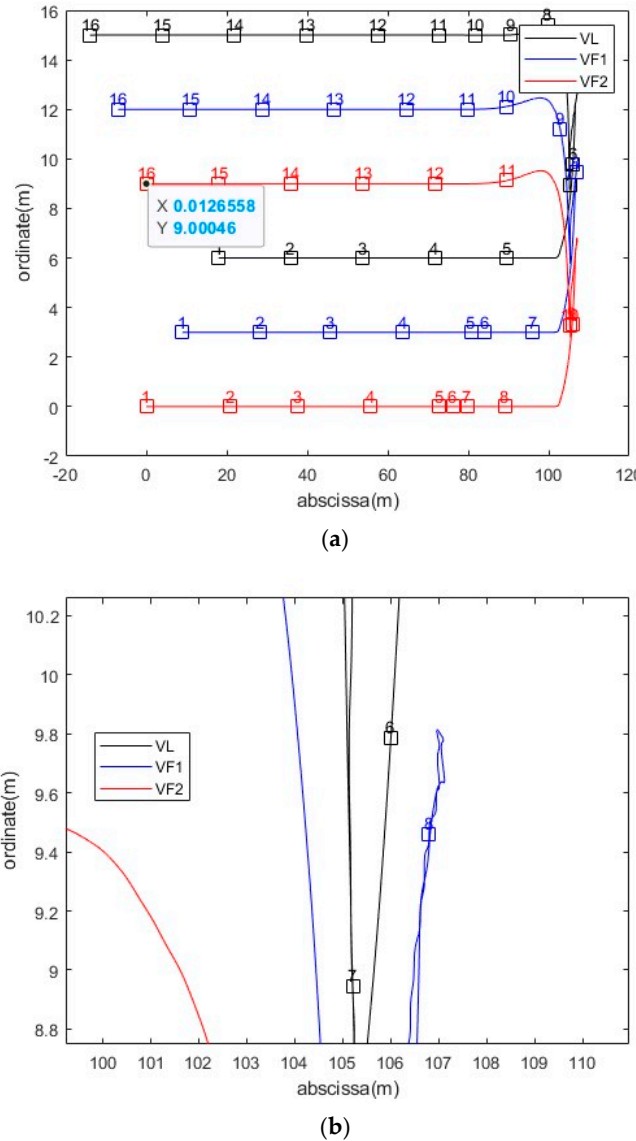

(**a**)

(**b**)

**Figure 17.** T-turn cooperative steering trajectory diagram: (**a**) global trajectory and (**b**) local trajectory.

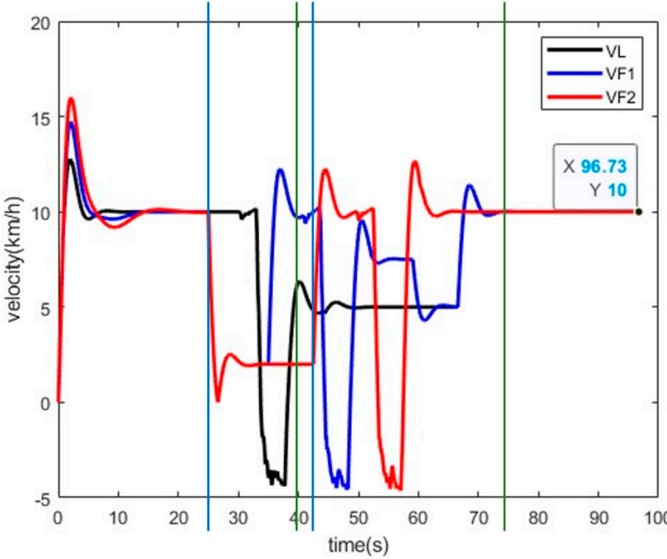

**Figure 18.** T-type cooperative steering longitudinal speed diagram.

Figure 19 illustrates the relationship between the spacing of the three agricultural machines during the T-turn collaborative turning process, and their correlation with the minimum safety distance, corresponding to the red dashed line.

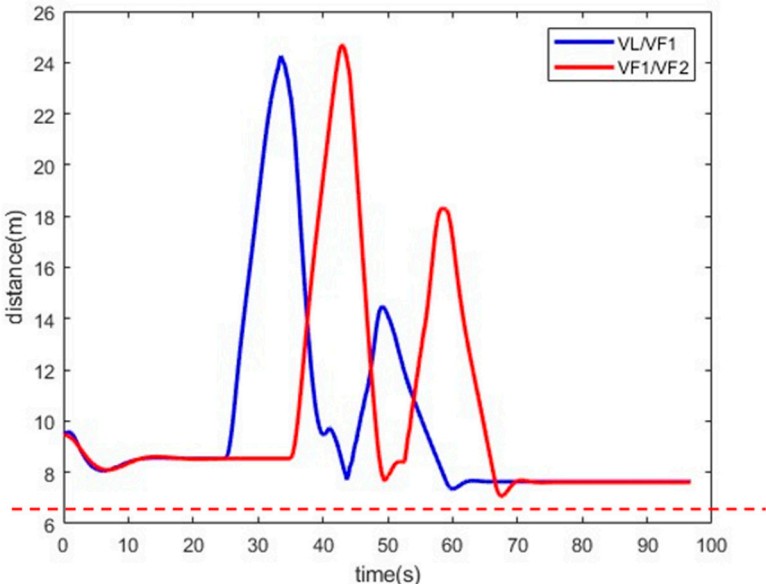

**Figure 19.** T-type cooperative steering agricultural vehicle distance diagram.

In addition, under the same scenario as the T-turn cooperative turning, a simulation of sequential cooperative steering control is conducted, involving a straight-line segment of 200 m at a forward speed of 10 km/h and a reverse speed of 5 km/h. The steering controller halts the rear machine once the front machine initiates steering and waits until the front vehicle has fully completed the turn. The rear machine begins its turn after the front machine reaches point T4 in Figure 2b. The simulation results are illustrated in Figures 20 and 21. The squares and numerals within the figure carry the same meaning as in the T-turn cooperative steering scenario, representing the location of the machines at various time intervals and the sequential order of these moments, respectively.

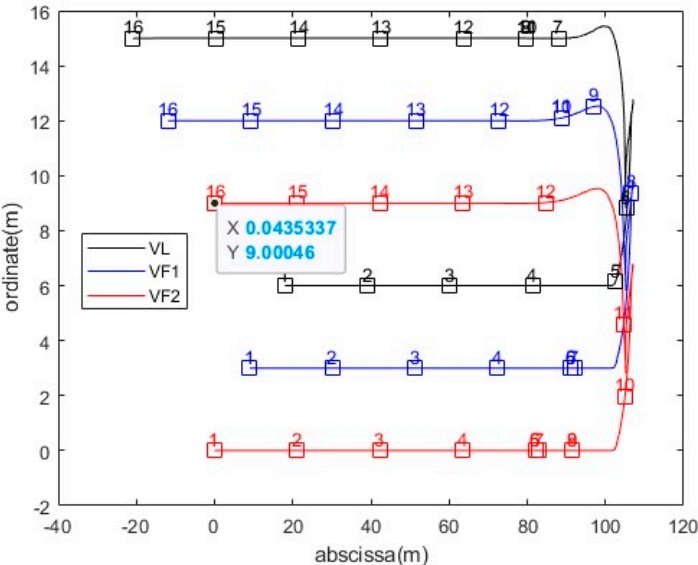

**Figure 20.** T-turn cooperative steering comparison method trajectory diagram.

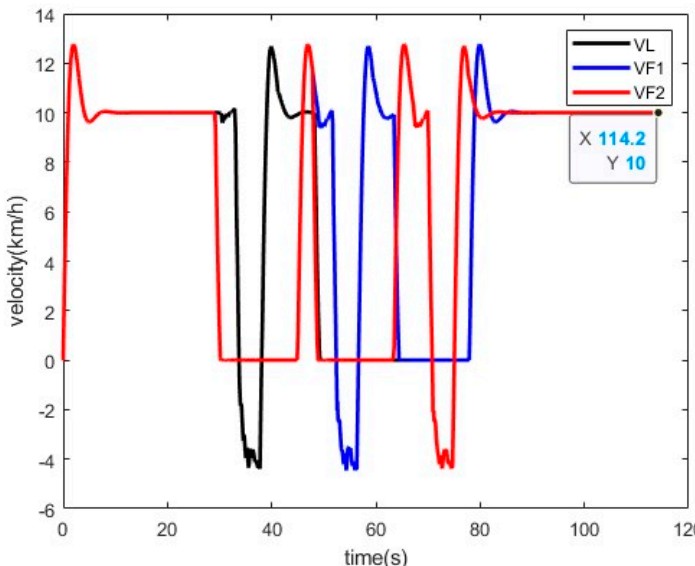

**Figure 21.** T-turn cooperative steering comparison method longitudinal speed diagram.

The initial coordinates for the comparative methods are the same as those mentioned earlier for the T-turn steering. Similarly, when VF2 completes the operational task, reaching coordinates (0, 9), it is considered that the three agricultural machines have completed the task, and the simulation is terminated, corresponding to a time of 114.2 s. Therefore, it can be considered that the control method proposed in this paper, compared to the sequential control method, reduces the operation time per 200 m by 17.47 s, thereby improving efficiency by 15.29%.

## 5. Conclusions

Addressing the issue of unnecessary stopping and waiting, as well as reduced operational efficiency due to the sequential control in existing multi-machine cooperative headland steering, this paper proposes a novel method for multi-machine cooperative straight-line and headland steering that accounts for both U-turn and T-turn cooperative maneuvers. The method comprises the design of a path-following controller, a formation-keeping controller, and a headland cooperative steering controller. The simulation results indicate that during U-turn cooperative steering, three agricultural machines can successfully navigate a pre-planned path and maintain formation while collaborating and steering without any risk of collision. In the case of T-turn cooperative steering, the three machines can complete the process without collision risks, unnecessary stopping, or waiting. Compared to sequential control, the operational time is reduced by 17.47 s per 200 m, enhancing efficiency by 15.29%.

However, this paper has some apparent limitations: (1) The kinematic model of agricultural machinery is simplified to a two-wheel model with front-wheel steering, without considering actual position feedback. (2) The cooperative steering control is only addressed for the same model of agricultural machines, neglecting coordinated steering control for machines with different turning radii or operating widths. Further discussions and investigations addressing these limitations will be pursued in subsequent research endeavors.

**Author Contributions:** Conceptualization, W.Z. and Y.Z.; methodology, W.Z.; software, W.Z.; validation, W.Z., Y.Z. and W.K.; formal analysis, P.J.; investigation, Y.Z.; resources, F.J.; data curation, W.Z.; writing—original draft preparation, W.Z.; writing—review and editing, Y.Z.; visualization, W.Z.; supervision, Y.Z.; project administration, W.K. and F.J.; funding acquisition, W.K. All authors have read and agreed to the published version of the manuscript.

**Funding:** This research was sponsored by the National Key Research and Development Program of China (2022YFB2503203) and the Beijing Nova Program (20220484040).

**Data Availability Statement:** The original contributions presented in the study are included in the article, further inquiries can be directed to the corresponding author.

**Acknowledgments:** The authors would like to thank the National Key Research and Development Program of China (2022YFB2503203) and the Beijing Nova Program (20220484040).

**Conflicts of Interest:** The authors declare no conflicts of interest.

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
