# Peer review of "A Versatile Control Method for Multi-Agricultural Machine Cooperative Steering Applicable to Two Steering Modes"

_wevj, doi:10.3390/wevj15040126_

Round 1
Reviewer 1 Report
Comments and Suggestions for Authors
I am overall not happy with the structure of the paper. Below are my comments. First, the author needs to dig deeply into the literature review section, which is missing in the paper. Second, the contribution of the paper is unclear. Third, it is not clear what the questions are to conduct the study. Fourth, the author should just bring the methodology into one section and not divide it into separate ones. Lastly, the result was not presented in detail. Please address all of them and return the paper.
Comments on the Quality of English LanguageNeeds improvement.
Reviewer 2 Report
Comments and Suggestions for Authors
The paper is well written and the topic is also interesting. I only have the following comments:
1. Please remove the blurring figures with the better ones.
2. Please add one table showing the pseudo-code or the algorithm of the proposed method.
Reviewer 3 Report
Comments and Suggestions for Authors
Review Report
- A summary (one short paragraph) outlining the aim of the paper and its main contributions.
Paper name: “A Cooperative Steering Control Method for Multiple Agricultural Machines with Two Steering Modes Are Considered” by Weizhen Zhu, Yuhao Zhang, Weiwei Kong, Fachao Jiang and Pengxiao Ji.
In this paper, a novel cooperative control approach for multiple agricultural machines is proposed, considering two typical steering modes of farm machinery. Based on the farm machines' established lateral and longitudinal kinematics models, the method includes a path-tracking controller designed using the Pure Pursuit and Stanley algorithms, a formation-keeping controller based on PID control, and a T-turn cooperative steering controller based on a problem-solving approach. Simulation results indicate that the proposed control framework and methodology can effectively ensure no collision risk during the U-turn and T-turn cooperative steering processes for three farm machines, eliminating stopping in T-turn, enhancing safety, and improving fuel economy.
Broad comments highlighting areas of strength and weakness. These comments should be specific enough for authors to be able to respond.
The paper is well structured, containing an Introduction with a review of the state-of-the-art in electrohydraulic servo systems, in which they are presented in detail, but still focused on the subject, 22 referenced works. Section 1 describes the Problem description. Section 2 contains the kinematic model of the Multiple Agricultural Machine Cooperative Control Architecture and Agricultural Machine. Section 3 presents the Multiple Agricultural Machine Cooperative Controller Design. The 4 Section contains the Simulation Validation of the system with the two classical challenges of the U-turn and T-turn. The conclusions follow the article’s line, which is consistent in relating to the subject.
1. Can you find a better title?
2. Please follow the chapter numbers as in the template, starting with 1 Introduction, not from 0.
3. References are more visible presenting them inline [1], instead of [1].
4. Please expand the introduction for presenting all the references [12-20], individually. MDPI has no page limits to cut the content. Thank you.
5. Figure 3 - Please align vertical text at 90Ëš counterclockwise.
6. Figure 7. Please use English.
7. Is any reference available for the mentioned Separating Axis Theorem on line 205? That paper has other authors and other content, being an OBB analysis.
8. Equation (12) - specify measurement units if 1.5.
9. For agricultural vehicles, the direction is not set solely by the front wheel orientation. Please write in the discussion that there is such a limitation of this method, which has not been considered a position feedback.
10. Speed from Figure 14 and Figure 17 is greater than the maximum speed of the tractor considered, as in Table 1. Is this an algorithm for over-powered machines, only?
11. Discuss more of the initial position in Figures 11 a and b, and the initial positions in Figures 15 and 18.
12. Please present more data to support the conclusions from line 414, as those numbers appear from nowhere, with no equation, no graph, and there is no observable result. This is the greatest issue of this article.
13. For the references, provide DOI where available. If possible, refer to the papers written in English, and published in open-access, where available. Check all the references. For instance, [25] have different authors than the presented ones.
Specific comments, refer to line numbers, tables, or figures. Reviewers need not comment on formatting issues that do not obscure the meaning of the paper, as editors will address these.
An ambitious article, but with several inconsistencies and a major problem regarding the conclusions that are not supported by the presented material. The good part is that it can be improved but with more work.
Round 2
Reviewer 1 Report
Comments and Suggestions for Authors
The author replied to my comments. So, the paper can be accepted.
Comments on the Quality of English LanguageNothing.
Reviewer 2 Report
Comments and Suggestions for Authors
Please increase the font size of Figures 3, 4, and 8
Figures 14, 15, 16, 17, 18, 19, 20, and 21 are still blurring please replace them with the better one.
Reviewer 3 Report
Comments and Suggestions for Authors
I must thank the authors for the revised version of the manuscript, where all previous observations were adequately treated.
For better readability, the figures can be magnified.
Paper is good for publishing.
